# The MHC Class-I Transactivator NLRC5: Implications to Cancer Immunology and Potential Applications to Cancer Immunotherapy

**DOI:** 10.3390/ijms22041964

**Published:** 2021-02-17

**Authors:** Akhil Shukla, Maryse Cloutier, Madanraj Appiya Santharam, Sheela Ramanathan, Subburaj Ilangumaran

**Affiliations:** 1Department of Immunology and Cell Biology, Faculty of Medicine and Health Sciences, Université de Sherbrooke, Sherbrooke, QC J1H 5N4, Canada; Akhil.Shukla@USherbrooke.ca (A.S.); maryse.cloutier@usherbrooke.ca (M.C.); Madanraj.Appiya.Santharam@USherbrooke.ca (M.A.S.); Sheela.Ramanathan@USherbrooke.ca (S.R.); 2CRCHUS, Centre Hospitalier de l’Université de Sherbrooke, Sherbrooke, QC J1H5N4, Canada

**Keywords:** NLRC5, MHC-I, cancer immunotherapy

## Abstract

The immune system constantly monitors the emergence of cancerous cells and eliminates them. CD8^+^ cytotoxic T lymphocytes (CTLs), which kill tumor cells and provide antitumor immunity, select their targets by recognizing tumor antigenic peptides presented by MHC class-I (MHC-I) molecules. Cancer cells circumvent immune surveillance using diverse strategies. A key mechanism of cancer immune evasion is downregulation of MHC-I and key proteins of the antigen processing and presentation machinery (APM). Even though impaired MHC-I expression in cancers is well-known, reversing the MHC-I defects remains the least advanced area of tumor immunology. The discoveries that NLRC5 is the key transcriptional activator of MHC-I and APM genes, and genetic lesions and epigenetic modifications of *NLRC5* are the most common cause of MHC-I defects in cancers, have raised the hopes for restoring MHC-I expression. Here, we provide an overview of cancer immunity mediated by CD8^+^ T cells and the functions of NLRC5 in MHC-I antigen presentation pathways. We describe the impressive advances made in understanding the regulation of NLRC5 expression, the data supporting the antitumor functions of NLRC5 and a few reports that argue for a pro-tumorigenic role. Finally, we explore the possible avenues of exploiting NLRC5 for cancer immunotherapy.

## 1. Introduction

The concept of natural anti-cancer immunity faced decades of skepticism before receiving formal acceptance with the recognition of ‘cancer immune evasion’ as one of the hallmarks of cancer in the updated treatise of Hanahan and Weinberg [1,2]. On the heels of this recognition came the ‘breakthrough of the year’ in 2013 from the pioneering works of Tasuku Honjo and Jim Allison on immune checkpoint blockers that launched the era of ‘cancer immunotherapy’ which has become so successful that the term has entered the mainstream lexicon [3,4]. In fact, the foundation for cancer immunology and immunotherapy was laid more than a century ago by William Coley, reporting in 1893 the ability of toxic products from pathogenic streptococci to cause tumor regression despite adverse and even fatal side effects [5]. Even though this work faded into oblivion with the emergence of effective cancer treatments such as radiation therapy and surgery, the ability of bacterial products to boost the body’s defense against cancer is still supported by the use of *Mycobacterium bovis* BCG to treat bladder cancer [6,7]. The idea that immune cells might be involved in the body’s fight against cancer, originally suggested by Paul Ehrlich in 1909, was rekindled fifty years later when Lewis Thomas and Frank Macfarlane Burnet put forth the concept of ‘immunological surveillance’ against newly arising neoplastic cells bearing mutations [8]. This concept was experimentally proven by Robert Schreiber and colleagues another forty years later [9]. Meanwhile, understanding of the cellular immune mechanisms paved the way for using the T cell growth factor interleukin-2 (IL-2) to stimulate anti-cancer CD8^+^ cytotoxic T lymphocytes (CTLs) in cancer patients and to expand these CTLs in vitro for the purpose of adoptive cell therapy (ACT) [10,11]. Even though these cancer immunotherapy approaches have been recently shadowed by the tremendous success of immune checkpoint inhibitors (ICI), IL-2 therapy is still being used to treat certain cancers such as renal cell carcinoma [12]. Similarly, the knowhow developed around ACT is applicable to personalized cancer immunotherapy using chimeric antigen receptor bearing T (CAR-T) cells [13].

Cancer immune surveillance begins with the detection of potentially neoplastic cells by naïve T lymphocytes via recognition of non-self antigenic epitopes, which are sufficiently different from ‘self’ epitopes for which T cells were educated to be tolerant during development within the thymus. Ensuing activation of these T cells, their expansion and killing of target cells that express ‘non-self’ antigens results in the elimination of potentially neoplastic clones, preventing them from growing into tumors. Essentially, the immune system acts as a ‘cell-extrinsic’ tumor suppressor analogous to ‘cell-intrinsic’ tumor suppressors such as p53 to maintain ‘self’ by eliminating the ‘non-self’ [14]. Genetic events that facilitate aggressive growth may permit tumors to select neoplastic clones that no longer express the ‘immunogenic’ tumor antigens in order to overcome cancer immune surveillance. Iteration of these processes enables tumors get past through stages of ‘elimination’ by the immune system, ‘equilibrium’ with the immune system and ‘escape/evasion’ from immune detection—the ‘three Es of cancer immunoediting’—first proposed by Robert Schreiber [9,15]. At the same time, by studying different murine tumors, Zinkernagel and colleagues demonstrated that activation of antitumor immunity can be quite variable depending on several factors such as the strength of the antigenic epitope, presence or absence of inflammation and the ability to hide within lymph nodes where T cell activation occurs. In addition, certain tumors avoid activating T cells either by tolerizing the immune system or by resisting immune recognition [16,17]. It is now well established that cancer cells exploit a myriad of cell-intrinsic and cell-extrinsic strategies within the tumor microenvironment and in lymph nodes to prevent both activation of T cells against the ‘non-self’ antigens and to dampen the effectiveness of activated antitumor CTLs [18,19,20]. This knowledge has provided the blueprint to devise diverse strategies aimed at reactivating the immune system and improving its fight against cancer (reviewed in [21]). Current cancer immunotherapy approaches are predominantly aimed at (i) stimulating anti-cancer T cells (through identification of tumor antigens for personalized vaccines, (ii) inducing immunogenic cell death of tumors (chemotherapeutic agents, killing by oncolytic virus), (iii) achieving efficient activation of antitumor T lymphocytes (via blocking checkpoints, inhibiting immune suppressive cells) and (iv) introducing tumor-reactive CTLs (antitumor CTLs expanded in vitro, engineered CAR-T cells targeting specific tumor antigens), either individually or in different combinations. For all these strategies to be successful the cancer cells must remain susceptible to attack by CTLs. Cancer cells exploit this critical requirement by deploying a simple but effective strategy of hiding from CTLs. This strategy involves downmodulation of major histocompatibility class-I (MHC-I) molecules that are responsible for the presentation of cancer antigenic peptides to CTLs. Even though this phenomenon has been recognized for several decades in diverse cancers, little advance has been made so far in making hidden cancers visible to the immune system [22,23,24,25]. A breakthrough in this field came from the discovery of NLRC5 as the key transcriptional activator of genes coding for MHC-I and several key proteins involved antigen processing and the loading of antigenic peptides onto MHC-I molecules [26,27]. Subsequent reports that the expression of NLRC5 is widely compromised in many cancers by deletions, mutations and epigenetic repression highlighted the possibility of exploiting NLRC5 to reverse MHC-I expression defects in cancers [28]. Indeed, NLRC5 has been used in a preclinical model of murine melanoma to restore MHC-I expression, increase tumor immunogenicity and elicit protective antitumor immunity [29]. In the following sections, we briefly discuss cancer antigens, the antigen processing pathway that generates MHC-I binding peptides and the various defects of this pathway before describing in detail the biology of NLRC5, its implications in cancer immunogenicity and potential ways of exploiting NLRC5 to restore MHC-I expression in order to elicit antitumor immunity [26,27,28].

## 2. Cancer Immunogenicity and Tumor Antigenic Peptides

The ability of the immune system to eliminate ‘autologous’ (as opposed to possibly allogenic) cancer was first documented in 1943 using serial transplantation of methylcholanthrene-induced sarcomas in inbred C3H mice [30]. It took two decades for direct demonstration of the ability of lymphocytes isolated from immunized rats to cause tumor regression [31]. This was followed by studies documenting in vitro sensitization of immune lymphocytes by tumor cells, expansion of sensitized cells using IL-2 and the ability of the expanded lymphocytes to attack tumors [32,33]. Subsequently, transfection of tumor cells with genes coding for cytokines that promote antigen presentation (discussed later), activate T cells and APC or induce costimulatory molecules (IFNγ, IL-2, IL-4, IL-7, GM-CSF) were shown to increase tumor immunogenicity and elicit antitumor immunity [34,35,36,37]. Establishment of CTL clones from tumor-infiltrating lymphocytes (TILs) of human tumors, mainly melanoma, revealed that these clones recognized diverse tumor antigens [38,39,40]. Studies on CTLs reactive to murine tumors induced by viral antigens and CTLs reactive to human tumors established that MHC-I molecules are involved in tumor antigen recognition [41,42,43,44,45]. However, efforts to identify the tumor antigens remained challenging and the initial biochemical fractionation methods were largely unsuccessful [46]. Boon and others applied genetic engineering techniques to successfully identify genes coding for tumor antigens such as *MUC-1* and the *MAGE* (melanoma antigen gene) family proteins [47,48,49,50]. Meanwhile, work from Rammensee and others revealed that acid treatment of MHC-I molecules from tumor cells released 8–10 amino acids-long peptides that could activate antitumor CTLs [51,52,53,54]. Further works established that these tumor antigenic peptides are derived from mutant proteins (tumor-specific antigens, TSA) or overexpressed embryonic proteins (tumor-associated antigens, TAA) and are presented by MHC-I molecules. CD8^+^ T cells, which scan these peptides, recognize any ‘deviation from self’ and elicit antitumor immunity. Advances in genome sequencing and bioinformatics have now made it possible to detect tumor neoantigens in primary cancers and deduce the immunogenic peptides that can potentially be presented by MHC-I [55,56,57]. These methods are complemented by proteomic techniques to directly identify MHC-I bound cancer antigenic peptides that arise not only from protein coding sequences but also from non-coding sequences or non-linear coding sequences that are unlikely to be predicted by genome and transcriptome data. The latter include peptides generated from defective ribosomal products (DRiPs), non-coding RNA sequences and fusion peptides created by proteasome catalyzed peptide splicing, and they all could contribute to immune surveillance [58,59,60,61,62,63,64,65]. Prediction methods are being developed for identification of such de novo MHC-I bound peptides in silico [66,67].

## 3. Processing and Presentation of Cancer Antigenic Peptides by MHC-I

The T cell antigen receptors (TCR) of antitumor T lymphocytes recognize tumor antigenic peptides presented by MHC molecules [68]. Whereas CD4^+^ T cells recognize antigenic peptides generated from ‘endocytosed’ foreign proteins following cleavage by lysosomal proteases and loaded on to MHC class-II (MHC-II) molecules, CD8^+^ T cells recognize peptides generated by proteasomes from ‘endogenously synthesized’ proteins and presented on MHC-I molecules. The cellular machineries involved in the generation of antigenic peptides have been extensively reviewed in the literature [69,70], and only the key steps that are impacted by NLRC5 (discussed later) are briefly discussed here. Whereas the expression of MHC-II is restricted to professional antigen presenting cells (APC) such as dendritic cells (DC), all nucleated cells express MHC-I. Similar to CD4^+^ T cells, the initial activation of naïve CD8^+^ T cells requires TCR recognition of the antigenic peptide as well as the engagement of the costimulatory receptor CD28 with its ligands CD80 and CD86, which are upregulated on APCs by inflammatory stimuli [71,72]. This stringent requirement for CD8^+^ T cell activation by endogenous peptides of cancer cells, which do not express the classical costimulatory ligands, is achieved by antigen ‘cross presentation’. In this setting, dead tumor cells that are taken up by APCs, which are also activated by the inflammatory milieu of the tumor microenvironment, are degraded via the antigen processing and presentation machinery (APM) of the MHC-I pathway [73,74,75]. Antigen cross-presentation also enables provision of the CD4^+^ T cell help, which is needed for efficient activation of naïve CD8^+^ T cells against cancer antigens.

The MHC-I pathway of antigen presentation operates in all nucleated cells under normal conditions as an important mechanism of cellular protein homeostasis that uses proteasomes to recycle aged and defective proteins by cleaving them into peptides and amino acids [76]. In this process, peptides that can bind MHC-I find their way to the cell surface, where they are presented as ‘identity badges’ to assure the immune system that they belong to ‘self’ and that there is no pathogen invasion or deviation from ‘self’ has occurred. In order to generate antigenic peptides, misfolded proteins and those destined for routine turnover are modified by ubiquitin and targeted to the proteasome for degradation [76]. Recent findings show that proteasomes also generate de novo peptides by splicing non-contiguous sequences [63,64]. The proteasome is a huge multiprotein complex consisting of seven inner and seven outer rings of core proteins (coded by PSMA1-7 and PSMB1-7, respectively) that form a barrel within which proteolytic cleavage occurs [77,78]. This core proteasome is abutted by a layer of proteasome activator and regulatory subunits. The composition of certain core and regulatory subunits are subject to modulation by IFNγ, which generates the ‘immunoproteasome’ by substituting three core proteasomal subunits (β1, β2, β5) and increasing the expression of the proteasome activator PA28 to alter the cleavage specificity and enhance the generation of immunogenic peptides [77]. These peptides are transported across the endoplasmic reticulum (ER) by transport proteins called TAP1 and TAP2 (transporter associated with antigen processing) that form a heterodimeric complex on the ER membrane [79,80]. The ER-resident aminopeptidase ERAAP trims the peptide to optimal length to be accommodated by the peptide binding groove of MHC-I formed by the α1 and α2 domains of the MHC-I heavy chain complexed with β2 microglobulin (β2M). The naked MHC-Iα/β2M is bound by chaperones calreticulin and ERp57 and brought to the TAP by the ER-resident TAP-binding protein (TAPBP, also called Tapasin) to form the peptide loading complex (PLC) [81]. Peptide binding stabilizes MHC-I α/β2M complex, and in doing so, facilitates its transport to the cell surface. IFNγ upregulates the expression of several components of this pathway including MHC-I, β2M, TAP transporters and TAPBP, and thereby increases the antigen presentation functions of APCs and somatic cells [82,83,84]. Enhancing the MHC-I antigen processing pathway clearly underlies the ability of IFNγ to improve the antigenicity of poorly immunogenic tumors and enhance their ability to elicit antitumor immunity [84,85,86].

## 4. The Cancer-Immunity Cycle and ‘Immune Invisibility’ of Cancers

The development of antitumor immunity leading to the killing of cancer cells can be envisaged as a series of discrete steps as summarized by Chen and Mellman [21] (Figure 1). These steps involve (i) release of tumor antigens through spontaneous death of cancer cells caused by the scarcity of nutrients and oxygen, which is an inevitable consequence of rapid cell proliferation and inadequate vascular supply; (ii) capture of these cell fragments by APCs that migrate to the draining lymph nodes, process antigenic peptides and present them on MHC molecules; (iii) recognition of the MHC-I:tumor antigenic peptide complex and costimulatory ligands by naïve T lymphocytes, resulting in their activation, clonal expansion and differentiation toward effector CTLs; (iv) migration of effector CTLs via circulation and infiltration into tumors; (v) recognition of cancer cells presenting the same tumor antigenic peptides that activated the naïve CD8^+^ T cells; and (vi) killing of these cancer cells via lytic granules that releases more tumor antigens to stimulate naïve T cells. Reiteration of the above steps in a cyclical manner contributes to tumor elimination. However, cancer cells can modify the cellular and molecular components of the tumor microenvironment that can impact every step of the cancer-immunity cycle. These immune evasion strategies include (i) rendering cancer antigens less immunogenic, (ii) dampening the antigen presenting functions of APCs, (iii) decreasing their stimulatory capacity to naïve T cells and even rendering them inhibitory, (iv) interference with the infiltration of effector CTLs into tumors, (v) making cancer cells less visible to CTLs or (vi) simply preventing the effector cells from unleashing their killing functions [18,21,87]. In fact, advanced stage cancers often deploy several of these inhibitory pathways simultaneously. Advances in the understanding of these inhibitory mechanisms have identified molecules and strategies that can be used to thwart the inhibitory influence of cancers on the activation and antitumor functions of T cells [21]. For instance, blocking the immune checkpoint inhibition caused by CTLA-4 can boost the initial activation of naïve CD8 T cells, whereas blocking the inhibition mediated by PD-L1 permits cancer cell killing by activated CTLs and can also boost their reactivation [88,89]. However, any cancer immunotherapy strategy aimed at activating antitumor T cells or relieving the inhibitory signals imposed on activated cells is unlikely to be successful if cancer cells deploy immune evasion mechanisms that enable them to remain invisible to CTLs. Cancer cells can achieve such ‘immune invisibility’ by downmodulating the expression of MHC-I or the components of the APM that generate antigenic peptides essential for stabilizing cell surface MHC-I expression.

## 5. Defective MHC-I Expression in Cancers

Human cancers downregulate MHC-I molecules, known as human leukocyte antigens (HLA), to avoid destruction by antitumor CTLs [22,90,91]. The HLA class-I molecules include the classical (class-Ia) HLA-A, B and C molecules, which are highly polymorphic and ubiquitously expressed, and the non-classical (class-Ib) HLA-E, F, G, MICA (MHC-I chain related protein A) and MICB, which are less polymorphic and restricted in expression [92,93,94,95]. During cancer growth, cells with varying degrees of HLA-Ia expression defects may arise. These defects that occur in both primary and metastatic cancers may range from loss of one or more HLA-Ia gene alleles, loss of one or more HLA-I loci, loss of an HLA haplotype (an entire set of A, B, C loci) or total loss of HLA-I [22,96,97]. These defects have also been studied using mouse models to gain insight into the underlying mechanisms [23,86]. Whereas the selective loss of alleles and loci could arise from mutations and deletions of specific MHC-I gene loci, the total loss of MHC-I most likely arises from defective expression of β2M or any of the key components of the MHC-I antigen presentation pathway required to generate antigenic peptides and facilitate their binding to MHC [22,98,99,100]. Indeed, defective expression of TAP and TAPBP has been implicated in the loss of HLA-I expression in cancers, cancer cell lines and mouse models [101,102,103]. Defective MHC-I expression in cancers may also arise from impaired interferon signaling pathways [104,105]. The MHC-I defects have been shown to correlate with high tumor grading, disease progression, reduced survival and failure of CTL-based immunotherapies [91,103,106,107]. Notably, in melanoma patients undergoing immunotherapy, all regressing metastatic lesions expressed residual MHC-I while progressing metastases did not [107,108]. Efforts to understand the underlying mechanisms revealed that the MHC-I expression defects can arise from ‘soft’ reversible or ‘hard’ irreversible lesions [23,25,109,110]. Hard lesions arising from gene loss or structural mutations are less common than soft lesions arising from epigenetic modifications that impair gene expression. The latter is exemplified by restoration of MHC-I expression by IFNγ or drugs such as 5-azacytidine (5-Aza) and trichostatin-A that inhibit DNA methylation and histone deacetylation, respectively [84,111,112,113,114,115,116]. Interestingly, 5-Aza restored IFNγ-induced upregulation of MHC-I in a melanoma cell line [104]. MHC-I expression has also been associated with demethylation of TAP1 and TAP2 genes, suggesting that IFNγ functions at least partly as an epigenetic modifier of APM genes [117]. This idea is also supported by the enrichment of IFNγ-induced genes and MHC-I antigen presentation pathway genes among those induced by 5-Aza in breast, ovarian and colorectal cancer cell lines [118].

MHC-I expression may also be regulated at post-transcriptional level. For instance, the K3 family of ubiquitin ligases encoded by Kaposi’s sarcoma-associated herpes virus promote K63-linked polyubiquitination of MHC-I, resulting in its internalization and lysosomal degradation, as an immune evasion strategy [119,120,121,122,123,124]. The cellular orthologs of K3 family are the MARCH (Membrane-associated RING-CH-type finger) family E3 ligases, which mediate ubiquitination and lysosomal degradation of MHC-II molecules [125]. A recent study implicates MARCH-9 in the regulation of MHC-I molecules [126]. In colon and pancreatic cancer cells, oncogenic RAS signaling promotes autophagy that results in lysosomal degradation of MHC-I, leading to their immune escape [127,128,129].

## 6. Loss of NLRC5 Expression Frequently Underlies Reduced MHC-I Expression in Cancers

A breakthrough in understanding the regulation of MHC-I genes and their defective expression in cancers came from the work of Kobayashi group on the nucleotide-binding leucine-rich repeat-containing receptor (NLR) family protein NLRC5 [130]. The NLR family proteins function as innate immune receptors that recognize pathogen- and danger- associated molecular patterns in the cytosol [131,132]. Activation of certain NLR proteins (NOD1, NOD2) leads to activation of the NF-κB pathway, whereas others (NLRP1b, NLRP3, NLRC4) assemble into inflammasomes and activate caspase-1 to promote maturation and release of inflammatory cytokines IL-1β and IL-18 [131,132]. Another well-characterized NLR protein is NLRA, which does not activate NF-κB or inflammasome, but has been known since the early 90s as the MHC class-II transactivator (CIITA) due to its crucial requirement as a transcriptional coactivator of MHC-II genes [133,134]. Early studies using overexpression and knockdown approaches in cell lines implicated NLRC5 in negatively regulating inflammatory pathways by attenuating NF-κB activation and in lessening antiviral responses by inhibiting type-I IFN production [135,136,137,138]. However, several groups have reported that bone marrow-derived dendritic cells and macrophages established from *Nlrc5*^−/−^ mice failed to show any difference in NF-κB-dependent pro-inflammatory cytokine gene expression or protein production following exposure to LPS, viruses or bacteria [139,140,141,142]. Curiously, one of these studies showed that *Nlrc5*^−/−^ primary embryonic fibroblasts did show enhanced NF-κB signaling and pro-inflammatory cytokine secretion [142]. More recently, the Ferrero laboratory has recently shown that NLRC5-mediated attenuation of NF-κB signaling in macrophages is crucial to attenuate chronic inflammation of the gastric mucosa caused by *Helicobacter pyroli* [143]. It has been suggested that NLRC5 is an unlikely regulator of pro-inflammatory NF-κB signaling [144] although such a role can be envisaged in certain cells that may not harbor other robust control mechanisms or exhausted them. It is also noteworthy that NLRC5 may also contribute to inflammasome activation via co-operating with NLRP3 [145,146,147].

Intrigued by nuclear localization of NLRC5 and its structural similarity to CIITA, the Kobayashi group investigated modulation of gene expression by NLRC5 [130]. This seminal study revealed that NLRC5 upregulated a limited set of genes, notably those coding for MHC-I, β2M, APM (TAP1) and immunoproteasome components (LMP2/PSMB9/β1i, LMP7/PSMB8/β5i). This study also showed that NLRC5 is strongly induced by IFNγ and that NLRC5 targeting siRNA attenuated IFNγ-mediated upregulation of MHC-I, indicating that NLRC5 is a crucial mediator of IFNγ-stimulated upregulation of the MHC-I antigen presentation pathway. Studies on NLRC5 deficient mice, generated by several laboratories, revealed that NLRC5 is critical for basal and IFNγ-induced expression of MHC-I and APM genes, but is dispensable for the regulation of pathogen-induced inflammatory cytokines, although some studies reported increased TLR-stimulated inflammatory cytokine production [139,140,142,146,148,149]. Thus, in analogy to CIITA, NLRC5 is recognized as MHC class-I transactivator (CITA) [27]. Subsequent studies from the Kobayashi laboratory revealed that NLRC5 is inactivated in diverse cancers by a variety of genetic mechanisms including promoter methylation, copy number loss and mutations, and that the loss of NLRC5 expression correlates with reduced CTL activation and patient survival in several cancers including melanoma, bladder and cervical cancers [28]. This study also showed that the demethylating agent 5-Aza increased *NLRC5* gene expression, suggesting that the earlier findings on the effects 5-Aza in restoring MHC-I expression occurred at least partly via derepressing *NLRC5*.

## 7. Structure and Transcriptional Coactivator Function of NLRC5

NLRC5 is the largest member of the NLR protein family with a size of more than 204 kDa [150] (Figure 2A). The *NLRC5* gene is highly conserved in vertebrates, with mouse and human NLRC5 containing 1915 and 1866 amino acids, respectively [135,151]. The NLR proteins display a tripartite domain architecture with a variable N-terminal protein interaction domain, a conserved central nucleotide-binding oligomerization domain (NBD/NOD) called NACHT (named after NAIP, CIITA, HET-E, and TP-1 proteins) and leucine-rich repeats (LRR) in the C-terminus that sense molecular patterns and vary in number in different NLR proteins [132,152]. Both NLRA/CIITA and NLRC5/CITA possess the central NACHT domain but vary in their N- and C-termini. Whereas CIITA harbors a caspase recruitment domain (CARD) and a trans-activator domain, the N-terminus of NLRC5 carries an atypical CARD domain [153]. CIITA carries four LRRs whereas NLRC5 harbors twenty LRRs although shorter isoforms containing fewer LRRs have been reported [27,137].

Unlike most NLR proteins that operate within the cytosol, the transactivation function of CIITA and NLRC5 requires them to shuttle between the cytosol and the nucleus [27,130]. The NBD/NACHT domain consists of Walker A and Walker B motifs that bind and hydrolyze nucleotide triphosphate (GTP/ATP), respectively. A nuclear localization signal (NLS) is located upstream of the NACHT domain. Both Walker A motif and NLS of NLRC5 are critical for nuclear localization, promoter binding and MHC-I gene induction [130,154,155]. The promoters of MHC-I and APM genes share the *cis*-regulatory elements of the MHC-II gene promoter namely, the W/S, X1, X2 & Y boxes (collectively referred to as the SXY regulatory module) that recruit DNA binding factors [156,157] (Figure 2B). The X1 box binds to a heteromeric DNA binding complex RFX composed of RFX5, RFX-associated protein (RFXAP), and RFX-associated ankyrin-containing protein (RFXANK/RFXB), whereas the X2 box is bound by cAMP-responsive element binding protein (CREB1) or the activating transcription factor 1 (ATF1), and the Y box binds to the Nuclear Factor-Y (NF-Y) composed of NF-YA, NF-YB and NF-YC subunits [157]. These factors are ubiquitously expressed and constitutively bound to the promoters of MHC genes and constitute the ‘transcriptional enhanceosome’. Steimle et al., discovered that CIITA (NLRA) is the essential co-activator of the MHC-II enhanceosome but is dispensable for MHC-I gene expression [133]. The work of Meissner et al., showed that co-activation of the MHC-I promoter is mediated by NLRC5, which is recognized as the *bona fide* CITA [130]. Both CIITA and CITA do not bind DNA but interact with the enhanceosome factors to activate MHC gene transcription. Despite the conserved nature of the SXY module and the transcription factors that bind to these motifs, NLRC5 does not influence MHC-II expression [130,158]. By comparing the genes regulated by NLRC5 and CIITA in cells derived from mice lacking NLRC5, CIITA or both, Ludigs et al. showed that these two related transcriptional coactivators activate distinct sets of genes and attributed this specificity to significant sequence divergence within the consensus SXY module [158]. Specifically, in contrast to the tight spacing constraints between S, X and Y motifs within the CIITA binding sites, the X-Y spacing is more relaxed within the NLRC5-binding sites. More importantly, the consensus S motif sequence of the NLRC5 binding sites was found to be quite distinct from that of CIITA binding sites and play a key role in determining the specificity. However, increasing the abundance of CIITA can lead to promiscuous activation of MHC-I promoter constructs [158].

Similar to MHC-II genes, MHC-I genes are induced by IFNγ, and CIITA can promote IFNγ-induced MHC-I expression [159,160]. This is achieved through the IFNγ-induced interferon regulatory factor-1 (IRF1) binding to the IFN-stimulated response element (ISRE) [159,161]. NLRC5, in addition to activating the enhanceosome, also co-operates with IRF1 [26]. The MHC-I promoters may also harbor one or two NF-κB binding motifs within the enhancer A site that account for the TNFα-mediated MHC-I expression [162,163]. Some MHC-I gene promoters carry binding sites for additional transcription factors such as Sp-1 [162,164]. Even though the promoters of most of MHC-I, β2M and APM (TAP1, PSMB9/LMP2) genes share the SXY module, there is considerable variation in the presence and number of the auxiliary ISRE and enhancer A elements [157]. These additional regulatory elements, co-operating with CIITA may account for the residual MHC-I expression observed in NLRC5 deficient mice. CIITA promotes MHC-II expression by recruiting various chromatin modifying factors histone acetyltransferases, deacetylases and methyltransferases [165]. ChIP assays on NLRC5-deficient hematopoietic cells indicate that NLRC5 relieves the silencing effect of histone methylation (H3K27me3) at MHC-I promoters [140].

## 8. Role of NLRC5 in MHC-I Expression, CD8^+^ T Cell Development and Functions

Even though different studies reported highly variable levels of NLRC5 expression in different tissues, all studies show elevated expression in hematopoietic cells and tissues [135,136,149,151,154]. CD8^+^ and CD4^+^ T lymphocytes in mouse and human show constitutively high NLRC5 expression. B lymphocytes, natural killer (NK) cells and NK-T cells also show high NLRC5 expression, whereas macrophages and dendritic cells show intermediate levels. Genetic ablation of the *Nlrc5* gene in mice results in drastic reduction of surface MHC-I protein expression in lymphoid cells (thymic and peripheral CD4^+^ and CD8^+^ T lymphocytes, NK, NKT and γδ T cells), intermediate decrease in B cells and a mild reduction in dendritic cells and macrophages [140,149,166,167]. Unlike CIITA deficiency, which results in impaired maturation of double positive thymocytes to CD4 single positive T cells, *Nlrc5* gene deletion does not affect the generation of mature CD8 single positive cells despite causing notable reduction in MHC-I protein expression in hematopoietic cells [146,149,168,169]. As the complete loss of MHC-I abolishes CD8^+^ T cell maturation, the lack of appreciable impact of NLRC5 deficiency on CD8^+^ T cell development could be explained by the high constitutive expression of MHC-I in thymic epithelial cells and its moderate reduction by NLRC5 deficiency [158,170]. It is noteworthy that skin epithelial cells display only moderate levels of constitutive MHC-I expression despite very high levels of *Nlrc5* transcripts, suggesting additional regulation of MHC-I gene transactivation [170].

Whereas constitutive MHC-I expression is dependent on occupation of the SXY module by transcription factors and their co-activation by NLRC5 [158], induced upregulation of MHC-I expression is mediated by the synergistic effect of activating additional *cis*-regulatory elements (ISRE and enhancer A) and the induction of NLRC5 itself. IFNγ strongly induces the expression of NLRC5 in both hematopoietic and non-hematopoietic cells, whereas type-I IFNs (IFN-I: IFNα, IFNβ) cause moderate upregulation. Agents that induce IFN-I such as virus infection and TLR ligands such as polyinosinic-polycytidylic acid (poly I:C; TLR3), lipopolysaccharide (LPS; TLR4) and CpG oligonucleotides (TLR9) also cause moderate upregulation of NLRC5 [136,142,149,151,154]. IFN stimulation results in the formation of STAT1 homodimers that binds the gamma activated sequence (GAS) at *IRF1* and *NLRC5* gene promoters to induce their expression, resulting in their synergy at the MHC-I gene promoters [171]. IFN and STAT1 activation are also implicated in the upregulation of NLRC5 in activated CD4^+^ and CD8^+^ T lymphocytes [149]. During viral infection, IFN-I protects activated CD8^+^ T cells from NK cell-mediated killing by upregulating classical and non-classical MHC-I molecules [172]. The Guarda laboratory showed that NLRC5 is the crucial mediator of IFN-I mediated upregulation of MHC-I in CD4^+^ and CD8^+^ T cells during inflammatory conditions and viral infection and this upregulation is crucial to inhibit their killing by NK cells [173].

The impact of NLRC5 on the ability of APCs to activate CD8^+^ T cells has been investigated in several studies using NLRC5-deficient mice. Bone marrow-derived macrophages (BMDM) from NLRC5-deficient mice pulsed with the SINFEKKL peptide induced proliferation of OT-I TCR transgenic CD8^+^ T cells as efficiently as control BMDCs [149]. However, NLRC5 deficient T cells served as poor CTL targets for OT-I cells in vitro. In contrast to BMDCs, NLRC5-deficient B cells pulsed with the same peptide were less efficient than wildtype B cells in inducing proliferation of OT-1 cells [148]. Even though this difference could be attributed to lower levels of MHC-I (H-2K) in resting B cells than in LPS-stimulated BMDM, another study reported that LPS-stimulated bone-marrow-derived dendritic cells (BMDC) from NLRC5-deficient mice were also less efficient in a similar experimental setting [146]. Whether the observed differences in the requirement for NLRC5 to present exogenously added peptide reflects the amount of surface MHC-I in the different cell types needs to be determined under controlled conditions using cells derived from same mice. Nevertheless, as NLRC5 regulates not only MHC-I genes but also APM genes, whether NLRC5 is needed for the physiologically relevant CD8^+^ T cell activation mediated by endogenously synthesized and cross-presented antigens has been addressed in vitro and in vivo. The Guarda laboratory expressed GFP-tagged SIINFEKL in BMDC and found no appreciable difference in OT-I cell activation despite reduced presentation of endogenous peptide [174]. NLRC5 deficiency reduced the number of IFNγ-producing CD8^+^ T cell numbers in the spleen and liver following intravenous infection with *Listeria monocytogenes*, accompanied by increased bacterial load [146,148]. Kanneganti and colleagues also observed reduced antigen-specific CD8^+^ T cell numbers in the lungs and draining lymph nodes of NLRC5-deficient mice following intranasal infection with influenza virus that was accompanied by increased viral titers, although the mice eventually recovered [166]. Even though the reduction in T cell numbers in these in vivo studies are generally attributed to impaired CD8^+^ T cell activation, these results could also be explained, at least partly, by the NK-cell mediated elimination of activated T cells as total NLRC5 knockout mice was used in these studies [173]. Sun et al. reported that mice lacking NLRC5 in CD11c^+^ DCs showed reduced CD8^+^ T cell numbers and altered immunodominance hierarchy in small intestinal lamina propria of NLRC5-deficient mice following oral Rota virus infection, indicating that NLRC5 impacts the antigen presentation functions of APCs [167]. Clearly further studies are needed to distinguish the requirement of NLRC5 in APCs for priming and cross-priming of naïve CD8^+^ T cells in the presence or absence of co-stimulatory signals, in activated CD8^+^ T cells for their survival, and in target cells such as cancer cells and virus-infected cells for rendering them susceptible to effector CD8^+^ T cells as this could facilitate cross-priming (Figure 1).

## 9. Induction of Butyrophilins by NLRC5 and γδ T Cell Activation

A recent report from the Guarda laboratory has implicated NLRC5 in activating γδ T cells via inducing the expression of butyrophilin (BTN) family proteins BTN3A1-3 [175]. γδ T cells express TCR with limited diversity that recognize self-MHC proteins in an innate-like fashion independently of their peptide cargo. These cells also recognize pathogen-encoded molecules and altered self-encoded molecules associated with disease states [176]. The human Vγ9Vδ2 TCR bearing cells recognize conformational changes in the extracellular domain of BTN3A1 resulting from the binding of low molecular mass phospho-antigens (pAg) to the intracellular B30.2 domain conserved among BTN and BTN-like proteins [177,178]. The pAgs are generated by deregulated mevalonate pathway in pathogen-infected and transformed mammalian cells. The BTN family is closely related to the B7 family of costimulatory proteins and are implicated in maintaining immune homeostasis by modulating T cell activity [178]. The human BTN family genes reside within the MHC locus on chromosome 6 and *Btn2a2* is regulated by CIITA via the SXY module [179]. Guarda and colleagues showed that human BTN3A genes (*BTN3A1*, *BTN3A2*, *BTN3A3*) also harbor the SXY module, and their expression positively correlates with NLRC5 [175]. NLRC5 was shown to bind the promoter region of BTN3A and induce its expression, suggesting a potential role for NLRC5 in immune homeostasis. Even though crosslinking BTN3A on T cells delivers an inhibitory signal, the impact of high expression of BTN3A in cancer cells on antitumor immunity is not yet clear [178]. Dang et al., showed that forced expression of NLRC5 in the human Burkitt lymphoma cell line Raji upregulated BTN3A expression and rendered them susceptible to killing by γδ T cells [175].

## 10. Regulation of NLRC5 Expression

As the NLRC5-mediated regulation of inflammation and MHC-I expression has potential application in cancer immunotherapy (discussed later), it is important to understand the underlying mechanisms regulating its expression. NLRC5 is strongly induced by IFNγ in different cell types, and to a lesser extent by type-I IFN [27] (Figure 3). In thymic epithelial cells, elevated basal NLRC5 expression is dependent on IFNλ [170]. Different studies have reported highly variable basal NLRC5 expression in different tissues [135,136,149,151,154], but the underlying regulatory mechanisms have not been thoroughly studied. The promoter region of NLRC5 (corresponding to position -1 to -1673 relative to the first exon), cloned from human genomic DNA, showed potent inducibility by IFNγ in reporter assays in HeLa S3 cells but was not induced by LPS [151]. On the other hand, LPS was a strong inducer of NLRC5 gene expression in murine primary macrophages and cell lines [135]. Kuenzel et al., predicted two STAT1 binding sites at -1336 and -452 relative to transcription stat site (TSS), with the distal site overlapping with a predicted NF-κB consensus sequence [151]. (Figure 3) These reports suggest variable chromatin accessibility of the NLRC5 promoter in different cell types and additional modulation by transcription factors. Indeed, epigenetic regulation by chromatin remodeling appears to be a key mechanism underlying differential NLRC5 expression. Hypermethylation of the NLRC5 gene promoter was reported to be the most common epigenetic mechanism associated with reduced MHC-I expression in human cancers and cell lines that could be reversed by 5-Aza treatment [28].

A recent CRISPR screen aimed at identifying genes responsible for low MHC-I expression in cancer cells identified the evolutionarily conserved polycomb repressive complex 2 (PRC2), which is known to modulate gene expression during embryonic development via histone methylation, and causes repressive histone methylation not only at MHC-I and APM gene promoters but also at the NLRC5 promoter [180]. This study implicated the lysine methyltransferase Enhancer of Zeste Homolog 2 (EZH2), a component of the PRC2, in mediating tri-methylation of histone H3 on lysine 27 (H3K27me3). This study also showed that pharmacological inhibition of EZH2 resulted in basal STAT1-independent restoration of MHC-I that was further upregulated by IFNγ stimulation. This could result from the combined effects of promoter de-repression and NLRC5-mediated transactivation. Inhibition of Embryonic Ectoderm Development (EED), a WD40 repeats-containing protein that potentiates the action of EZH2 within the PRC2 complex, also upregulated MHC-I. NLRC5 gene transcription is also inhibited by protein arginine methyltransferase 5 (PRMT5) which catalyzes methylation of arginine residues on several histones (H2AR3, H3R2, H3R8 and H4R3) [181,182]. Modulation of the epigenetic signature at the NLRC5 promoter is also implicated in STAT1-independent upregulation of NLRC5 and MHC-I genes in pancreatic cancer cells exposed to ionizing radiation, although an earlier report attributed radiation-induced MHC-I upregulation in breast cancer cell lines to secretion of IFNβ [183,184]. Increased CpG methylation of the NLRC5 and other IFN-I-responsive genes was reported in the genomic DNA of systemic lupus erythematosus patients that correlated with increased auto-antibody production, suggesting that epigenetic modulation of NLRC5 may also be influenced by systemic inflammation [185].Moreover, the chicken *NLRC5* gene was reported to harbor two CpG islands, one near the proximal core promoter that is unmethylated, and the second one encompassing an additional STAT1 binding site (distal to the STAT1-NF-κB site) that could be methylated [186]. Whether the mammalian NLRC5 promoter displays such additional regulatory elements and differential methylation patterns needs to be explored.

Recently, it was found that quiescent hair follicle stem cells and muscle stem cells downregulate the expression of MHC-I and APM genes by repressing NLRC5 in order to protect the stem cell pool from immune surveillance and destruction [187]. A previous study has shown that histone methylation represses the expression of MHC-I and APM genes in human embryonic and pluripotent stem cells [188]. It has also been well documented that cancer initiating cells of different cancers (melanoma, glioblastoma and lung cancer) express low MHC-I that would protect them from CTL-mediated destruction, facilitating generation of immune escape variants and causing cancer recurrence after therapy [189,190,191,192,193,194]. Hence, it appears that the downregulation of NLRC5 and MHC-I in cancer cells could be a part of the global genetic de-differentiation program that accompanies progressive cancer growth in order to maintain the cancer initiating cell population rather than an immune escape program. This idea is supported by the involvement of histone methylases PRC2 and PRMT5, which play important roles in modulating gene expression during embryonic development, in repressing the *NLRC5* gene promoter [180,181].

Transcription of NLRC5 is also subject to regulation by long non-coding RNA (lncRNA). A recent study showed that the lncRNA Arid2-IR inhibits NLRC5 gene expression and implicated this pathway in promoting renal inflammation [195]. The lncRNA Arid2-IR, located within the intronic region of AT-rich interactive domain 2 gene, was identified by RNAseq analysis of TGFβ-mediated inflammatory pathways in kidney diseases and renal fibrosis [196]. Arid2-IR, induced by Smad3, enhances IL-1β-induced NF-κB signaling without affecting TGFβ signaling. Zhou and colleagues discovered that Arid2-IR binds to the *NLRC5* promoter and prevents its transcription to maintain NLRC5 expression in medullary renal tubular epithelial cells at the basal level (Figure 3). However, during inflammation, inflammatory cytokines such as IL-1β induce the expression of filamin A, which traps Arid2-IR in the cytosol, relieving its suppressive effect in the nucleus and leading to increased NLRC5 expression and attenuation of NK-κB signaling (discussed later in this section). Zong et al., implicated another lncRNA in modulating *NLRC5* gene transcription [197]. Investigating the poor survival of glioma patients with high expression of the lncRNA SCAMP1 (Secretory carrier-associated membrane protein 1), this study showed that SCAMP-1 acts as a sponge for the miRNA miR499a-5p, thereby relieving its repressive effect on the LIM homeobox transcription factor 1, alpha (LMX1A) that binds the promoter region of the *NLRC5* gene (Figure 3) at multiple sites (-1168, -1452, -1734 relative to TSS). Whether differential expression of these lncRNAs could contribute to differential levels of basal NLRC5 expression in various tissues remains to be tested.

Interestingly, the Kufer laboratory cloned different NLRC5 isoforms from human leukocyte cDNA library, arising from alternate splicing and differing in the C-terminal region, raising the possibility that the NLRC5 isoforms may be destined for different functions within the leukocyte subpopulations [137]. However, unlike the CIITA isoforms that arise from different start codons and are tightly controlled by independent promoters in a cell-specific manner [198,199], the NLRC5 isoforms vary in their C-terminal LRRs. While it is conceivable that they may carry out different effector functions, whether their expression is also differentially regulated is not known.

Expression of NLRC5 is also regulated at the post-transcriptional and post-translational levels (Figure 3). The HIV-1 TAT protein was shown to upregulate the micro-RNA miR-34a in microglial cells, leading to downmodulation of NLRC5 and increased NF-κB activation that could contribute to neuronal inflammation [200]. On the other hand, miR-34a was reported to be downmodulated by human papilloma virus-16 (HPV-16) in cervical cancer cells, resulting in the upregulation of NLRC5 and consequent attenuation of NF-κB activation and pro-inflammatory cytokine production and virus persistence [201]. Zong et al., implicated miR-125b-5p in downmodulating NLRC5 expression [202]. This study showed that the lncRNA XIST (X-inactivation-specific transcript), upregulated in breast cancer cells, promotes cell proliferation, migration and invasion by increasing NLRC5 expression through sponging off miR-125b-5p. An obligate intracellular bacterium *Orientia tsutsugamushi*, which causes scrub typhus by infecting mononuclear and endothelial cells, was shown to attenuate NLRC5 expression in HeLa cells at the post-transcriptional level [203]. The blockade of NLRC5 protein expression by *O. tsutsugamushi* required bacterial protein synthesis and possibly host cell factors as inhibition of NLRC5 protein expression was reversible in a monocytic cell line THP-1 but not in endothelial cells. Molecular mechanisms underlying the inhibition of NLRC5 expression by *O. tsutsugamushi* at the post-transcriptional level remains to be elucidated.

NLRC5 attenuates the NF-κB pathway by interfering with the IKK complex, which is composed of IKKα and IKKβ kinases and the regulatory subunit NEMO (also called IKKγ). IKK-mediated phosphorylation leads to ubiquitination and proteasomal degradation of IκB, allowing NF-κB to translocate to the nucleus and induce gene transcription. Cui et al., have shown that, in LPS stimulated cells, NLRC5 inhibits IκB phosphorylation and degradation by binding to IKKα and IKKβ, thus preventing the IKK complex formation [135]. In an effort to elucidate the mechanisms underlying this regulation, the Cui group showed that TLR4 stimulation activates the TRAF2/6 complex, which ubiquitinates NLRC5 on Lys1178 residue, presumably leading to its degradation and release of IKKα and IKKβ to complex with IKKγ [204,205] (Figure 3) This study also showed that the ubiquitin-specific protease 14 (USP14) deubiquitinates NLRC5 to sustain the NLRC5-mediated inhibition of NF-κB activation. The NF-κB signaling pathway, which is activated in cancer cells by various stimuli including cytokines, growth factors, environmental stress and DNA damage, is implicated in cancer development, progression, metastasis and resistance to therapy [206,207].

A recent genome wide CRISPR screen aimed at identifying genes that regulate MHC expression, has identified several molecules in addition to the known transcription factors and promoter components [208]. Among them SUGT1 (human homologue of yeast SGT1), which has been previously reported to modulate NLR protein functions [209,210], appears to promote MHC-I expression by stabilizing NLRC5. SGT1 interacts with SKP1 within the SCF (Skp1-Cullin-F box) family of Ub ligase complex [211,212]. Further work is needed to determine how SUGT1 stabilizes NLRC5 protein. Overall, even though NLRC5 is repressed mainly by epigenetic mechanisms of DNA as well as histone methylation and histone deacetylation at its promoter, additional mechanisms involving miRNAs targeting NLRC5 and posttranslational modifications of NLRC5 protein could also contribute to reduced NLRC5 protein expression in cancers. The resulting downregulation of MHC-I expression and increased NF-κB activation can act in synergy to promote cancer growth.

## 11. Impact of NLRC5 on Antitumor Immunity

As the loss of MHC-I expression is a common immune escape mechanism in cancers and NLRC5 is the key transcriptional activator of MHC-I genes, our laboratory studied the impact of NLRC5 on antitumor immunity using the B16.F10 murine melanoma model [29]. The poorly immunogenic B16 cells are widely used in cancer immunotherapy studies [213]. These cells display negligible MHC-I expression that can be induced by IFNγ stimulation, indicating soft/reversible lesions in the MHC-I expression pathway [86]. We showed that stable expression of NLRC5 in B16 cells induces MHC-I and APM genes, upregulates cell surface MHC-I expression and promotes the processing and presentation of an endogenous tumor antigenic peptide from PMEL-1 (gp100) protein [29]. The NLRC5 expressing B16 cells showed reduced growth in immunocompetent C57BL/6 mice but not in Rag1-deficient mice, which lack mature T cells and B cells, indicating that NLRC5 expression promoted lymphocyte-mediated antitumor immunity. This was further supported by the reversal of reduced tumor growth in C57BL/6 mice depleted of CD8^+^ T lymphocytes. Moreover, immunization with irradiated NLRC5 expressing cells conferred protection against challenge with parental B16 cells. These findings indicate that MHC-I expression in APC is not sufficient, whereas MHC-I expression in tumor cells is necessary, to elicit protective antitumor immunity against MHC-I negative tumors [29]. Following this report, several other studies have documented similar findings (Table 1). In studying PRMT5-mediated inhibition of *Nlrc5* gene expression, Kim et al., confirmed that NLRC5 expression in B16 cells augmented MHC-I expression and reduced in vivo tumor growth [181]. Kalbasi et al. rendered B16 cells unresponsive to IFN by CRISPR-mediated deletion of *Jak1*, and showed that stable NLRC5 expression in these cells upregulated MHC-I and rendered them susceptible to immune elimination in C57BL/6 mice by adoptively transferred PMEL-1 TCR transgenic CD8^+^ T cells [214]. In pancreatic adenocarcinoma cells, the *Nlrc5* gene was shown to be induced by irradiation, resulting in increased susceptibility to anti-PD-L1 immune checkpoint inhibitor therapy [183]. The authors speculate that irradiation may alter epigenetic modification of the *Nlrc5* gene promoter. All these studies lend support to the notion that restoration of NLRC5 expression in cancer enhances MHC-I expression in tumor cells leading to efficient antitumor immune response and CTL-mediated killing of tumor cells.

## 12. Tumor Promoting Potential of NLRC5

Contrary to the studies highlighting the role of NLRC5 in reducing tumor growth by eliciting antitumor immunity, several reports suggest a potential role of NLRC5 in promoting cancer growth (Table 1). Li and colleagues observed elevated expression of NLRC5 protein in human hepatocellular carcinoma (HCC) specimens and in human HCC cell lines HepG2, SMMC-7721 and BEL-7402, and investigated the effects of NLRC5 knockdown and overexpression on cell growth [215]. NLRC5 overexpression promoted cell growth, motility and migration, and these effects were accompanied by elevated expression of β-catenin and downstream oncogenic signaling molecules. These NLRC5-mediated effects were reversed by the β-catenin inhibitor iCRT3. Knockdown of NLRC5 had similar effects and reduced the growth of HepG2 cells as a xenograft in BALB/c nude mice lacking adaptive immune cells. The same group reported similar NLRC5-mediated effects on clear cell renal cell carcinoma (ccRCC) cells [216]. Another recent study also implicated NLRC5 in promoting the growth of glioma cells by activating the Wnt/β-catenin pathway [197]. NLRC5 was also shown to activate the PI3K/AKT signaling pathway in human HCC and endometrial cancer cell lines and promote cell growth [217,218]. In esophageal squamous cell carcinoma (ESCC) and breast cancer cell lines, elevated NLRC5 expression was associated with lower levels of miR-4319 and miR-125b-5p, which reportedly target NLRC5 [202,219]. In ESCC cell lines, overexpression of NLRC5 promoted cell proliferation, colony formation and cell cycle progression. All the above reports implicating NLRC5 in promoting tumor growth in HCC, ccRCC, glioma and ESCC are based on in vitro studies on cell lines as well as in mice lacking a functional adaptive immune system, whereas NLRC5 inhibited growth of melanoma and PDAC cells in mice with a competent immune system. These discordant results may result from the antitumor immune response elicited by NLRC5 overcoming any growth stimulatory functions of NLRC5 in cancer cells. It is also possible that the different outcomes of NLRC5 may also depend on the tumor type. In this context, it is noteworthy that the Kobayashi group has observed elevated NLRC5 mRNA expression in liver, colon and brain cancer tissues within the TCGA study cohorts [28]. Even though this elevated NLRC5 expression was thought to result from high inflammatory conditions in these cancers, studies showing NLRC5 protein expression by immunohistochemistry reveal increased staining within epithelial cells [215,217]. The possibility that the antitumor versus pro-tumorigenic roles of NLRC5 may be influenced by the mutational load and the frequency of neoantigen generation of different cancers [56] also need to be considered. In this context, in colon cancers with microsatellite instability phenotype, NLRC5 mutations were reported to underlie a significant proportion of reduced MHC-I expression and contribute to immune evasion [220].

## 13. Restoring MHC-I Expression in Cancers

Despite extensive documentation of defects in MHC-I expression and antigen presentation in cancer cells, limited progress has been made in the efforts to reverse these defects [24,25,109,110]. As APM and β2M are necessary to generate stable MHC-I:peptide complex, introducing TAP1 and β2M genes using recombinant viral vectors was shown to restore surface MHC-I expression and elicit antitumor response in experimental tumor models [221,222,223,224]. While this approach may be useful if the MHC-I expression defect is caused by APM or β2M deficiency, it will not be useful if MHC-I genes were defective or their expression blocked.

IFNγ is a potent inducer of MHC-I and APM genes and promotes antigen processing by changing the proteasome constituents. IFNγ has been shown to restore MHC-I expression in cancer cells, suggesting its potential use to correct MHC-I defects [86]. However, cancers develop various defects to impair IFN signaling pathways [84,104,105], indicating that IFNγ therapy may not be useful in all cases. Moreover, IFNγ is a potent inducer of PD-L1, a ligand to the immune checkpoint receptor PD-1 [225,226,227,228]. Indeed, induction of PD-L1 by IFNγ produced by antitumor CTLs has been referred to as ‘adaptive immune suppression’ [229]. Furthermore, IFNγ can modulate many cellular components of the tumor microenvironment to induce promo-tumorigenic effects, limiting its potential use in cancer immunotherapy [230,231]. Curiously, IFNγ represses the expression of anti-inflammatory genes in macrophages via promoting recruitment of EZH2 and increasing H2K27me3 methylation, the same events that are implicated in repressing NLRC5 in stem cells [180,232]. Since IFNγ strongly induces NLRC5 gene expression, further studies in normal and cancer cells are needed to resolve the conundrum of STAT1-mediated gene activation and EZH2-mediated epigenetic repression of the same target genes.

IFNγ shares key elements of the JAK-STAT signaling pathway with type-I interferons that includes IFNα, IFNβ and several other members [228,233]. Whereas IFNγ receptors activate JAK1 and JAK2 and form STAT1 homodimers (also called gamma activated factor, GAF), IFN-I receptors activate JAK1 and TYK2, and form both STAT1 homodimers and STAT1:STAT2 heterodimers. Whereas STAT1 homodimers bind GAS sequences (present in IRFs and NLRC5 gene promoters), STAT1 homodimers and STAT1:STAT2 heterodimers interact with IRF9 to form the ISGF3 (interferon-stimulated gene factor 3) that binds the ISRE (IFN-stimulated response element, present in MHC-I and APM genes) to stimulate the expression of IFN-stimulated genes (ISG) and IRFs [228,233,234]. IFN-I, which confer resistance to viral infections by inhibiting transcription, degrading RNA and inhibiting protein translation, also results in cell growth inhibition and apoptosis via diverse mechanisms that can limit cancer progression [235,236]. IFN-I also promotes antitumor immunity by increasing MHC-I expression and APC functions [237,238]. Radiation and chemotherapeutic agents induce cancer cell autocrine and paracrine effects of IFN-I, which is crucial to mediate their antitumor activity [184,236,239,240]. However, the potential use of IFN-I as a cancer therapeutic has been limited by its systemic toxicity, although efforts are being made to overcome this limitation [238]. Besides, many cancers develop unresponsiveness to IFN-I in order to overcome its cytostatic effects by downmodulating its receptor chains, upregulating the negative regulators of the JAK-STAT pathway such as SOCS1, or reducing the expression of STAT1 [241,242]. Nonetheless, IFN-I unresponsiveness renders cancer cells susceptible to oncolytic viruses (OV), which are being exploited to kill cancer cells while sparing normal cells [243]. OV can also be armed with genes encoding immunostimulatory molecules such as tumor antigens, T cell costimulatory molecules and cytokines including IFNβ [244,245].

As discussed earlier, epigenetic repression of MHC-I expression and its possible reversal by epigenetic modifiers has been recognized for over two decades. Epigenetic modulation of gene expression is a fundamental process of chromatin remodeling that regulates embryonic development, cellular differentiation and adaptation to environmental challenges [246,247]. These mechanisms can contribute to the initiation and progression of cancer. Epigenetic modulation of gene expression can occur by DNA methylation of CpG island near promoters, histone methylation especially trimethylation of lysine on histone H3 (H3K27me3) and histone deacetylation. Inhibitors of DNA methyltransferases (DNMTi: 5-Aza, decitabine), histone deacetylases (HDACi: Vorinostat, Panobinostat, valproic acid, etc.) and histone methyltransferases (HMTi: EZH2 inhibitors -Tazemetostat) are undergoing clinical trials for cancer treatment with some being already approved [248]. 5-Aza and Trichostatin-A have been shown to derepress MHC-I and APM (TAP1) genes and restore MHC-I expression in human and mouse cancer cell lines as well as in xenograft models in mice [104,111,112,113,115,116,117,118,249]. The recent findings that NLRC5 gene expression itself is repressed by methylation of DNA and histones in cancer cells and primary cancers, and that inhibition of these repressive epigenetic mechanisms restores MHC-I expression raise the possibility of using epigenetic therapy to correct defective MHC-I expression in cancers [28,180,181]. However, certain limitations of epigenetic drugs may restrain their potential use to restore MHC-I via derepressing NLRC5. Epigenetic drugs have shown limited efficacy on solid tumors despite their remarkable success against hematopoietic cancers [250]. Another key challenge facing epigenome therapeutics is off- target effects [251,252]. Hence, in addition to using epigenetic modifiers, other approaches should be envisioned to restore MHC-I expression in cancers showing defective NLRC5 expression.

By comparing gene expression in MHC-I positive and negative variant of a mouse fibrosarcoma clone B9, Garcia-Lora and colleagues discovered that the loss of fragile histidine triad (*Fhit*) tumour suppressor gene expression was associated with the loss of MHC-I, β2M and APM gene expression [253]. Re-expression of *Fhit* restored the MHC-I expression in this clone. The same group recently reported that in MHC-I negative breast cancers the loss of NLRC5 is less frequent than the loss of FHIT, and that FHIT could be used to restore MHC-I expression [254]. As FHIT does not upregulate NLRC5 expression, the molecular pathways by which FHIT upregulates MHC-I remains to be determined [254].

## 14. NLRC5-Independent MHC-I Expression

A recent study shows that MHC-I expression in tumor cells can be induced independently of NLRC5 [214]. Clonal populations of B16 cells, rendered unresponsive to IFNs by the deletion of *Jak1* and also deleted of the *Nlrc5* gene, showed increased MHC-I expression following treatment with a nanoplexed version of polyI:C (BO-112). BO-112, which activated TLR3, resulted in NF-κB activation, and inhibition of NF-κB signaling abolished MHC-I expression. This study implies that increased NF-κB activity at the enhancer A site of MHC-I gene promoter may be sufficient to restore MHC-I expression without transactivating the NLRC5-dependent enhanceosome. Given the well documented role of NLRC5 in attenuating NF-κB signaling, the above study raises the possibility that MHC-I expression can be restored even in NLRC5 deficient cancers by inducing NF-κB signaling. However, these findings need to be replicated in other model systems. Moreover, it remains to be seen whether such an approach will lead to strong enough induction of antitumor immunity that can overcome the potentially harmful deregulation of NF-κB signaling. It is also noteworthy that IFNγ stimulation upregulated MHC-I in NLRC5-deficient T cells, B cells and macrophages, albeit to a lower extent than wild type controls [140]. Although CIITA may compensate for the loss of NLRC5 in B cells and macrophages after IFNγ stimulation, which can be tested in mice lacking both NLRC5 and CIITA, other unknown mechanisms likely underlie similar MHC-I induction in NLRC5 deficient T cells.

## 15. Role of NLRC5 in Cancer Immune Surveillance

The ability of NLRC5 to promote cancer immunogenicity by upregulating MHC-I and APM may also contribute to cancer immune surveillance. Whether tumor cell intrinsic NLRC5 expression is essential, or NLRC5 expression in APCs is sufficient, preventing the emergence of neoplastic clones remains to be addressed. Our findings on the B16 melanoma model indicate that NLRC5 expression in APCs alone is not sufficient to induce antitumor immunity, and that NLRC5 expression in tumor cells is crucial to elicit antitumor immune response, which was also effective against parental B16 cells that display reduced MHC-I expression [29]. It is likely that APCs acquire MHC-I bearing tumor antigenic peptides from NLRC5 expressing B16 cells, presumably via the process of ‘trogocytosis’ [255,256], and that such ‘cross-dressed’ APCs induce antitumor immune response more efficiently than APCs processing the antigens of parental B16 cells. This process may also occur for newly emerging neoplastic clones, contributing to cancer immune surveillance. It can be envisaged that newly arising immune escape variants may be able to generate effective immune response as long as they express NLRC5 and MHC-I. As ablation of the *Nlrc5* gene does not completely abolish MHC-I expression, especially in non-hematopoietic cells, it should be possible to test using NLRC5-deficient mice whether NLRC5 plays a crucial role in cancer immune surveillance and selection of immune escape variants.

NLRC5 not only transactivates classical MHC-Ia genes but also induces non-classical MHC-Ib genes, which may impact tumor immune surveillance. The non-classical MHC-I molecules are encoded by genes within H2-Q, H2-T and H2-M loci (each containing several genes) in mouse and non-orthologous *HLA-E*, -*F*, -*G*, *MICA* and *MICB* genes in human [94,257,258]. Most of the mouse MHC-Ib molecules and human HLA-E are broadly expressed whereas certain mouse MHC-Ib and human HLA-F and G show restricted expression pattern [94]. Most mouse MHC-Ib molecules and HLA-E present protein antigenic peptides [94,95]. These MHC-Ib molecules can activate unconventional CD8^+^ T cells bearing αβTCR to elicit rapid innate like effector functions that contribute to immune response against pathogens and neoplasms [95]. However, certain non-classical MHC-Ib molecules such as H2-T11/23 (also known as Qa-1) and HLA-E (the functional homologue of Qa-1) present a peptide derived from the leader sequences of classical MHC-Ia known as Qa-1 determinant modifier (Qdm), which delivers an ‘inhibitory signal’ to NK cells and CD8^+^ T cells upon engaging NKG2A receptor of the CD94/NKG2A family [259,260,261]. NK cell stimulation is determined by the net balance between inhibitory and activation receptor signaling. NK cells are activated by the failure to receive an inhibitory signal delivered by MHC-Ib molecules (missing self-recognition) as well as by signaling from activating receptors [262]. Thus, the interaction of NK cell inhibitory receptors with MHC-Ib serves to indirectly monitor defective MHC-Ia expression in cancer cells. However, cancer cells can evade NK cell-mediated killing by upregulating the expression of HLA-E that engage the NK cell inhibitory receptors and blocking this inhibitory signaling can boost antitumor NK and CTL responses [263,264,265]. Cancer cells also exploit HLA-G, which is expressed by fetal trophoblasts and maintain immune tolerance, for immune evasion [266,267].

The promoters of *HLA-E*, *F*, *G* genes harbor the SXY module and are induced by overexpressed NLRC5 in HEK293T cells [130,268]. In mice, NLRC5 deficiency was shown to decrease the expression of *H2-M3*, H-2T11/23 (*Qa1*) and H2-T18 (also known as *Tla*) [140,146,148]. By chromatin immunoprecipitation (ChIP) sequencing of NLRC5 bound genomic sequences in T cells (which show maximal decrease in MHC-Ia expression in NLRC5 deficient mice) Ludigs et al., showed that NLRC5 binds to the promoter regions of several MHC-Ib genes, including *H2-Q4*, *H2-Q6*, *H2-Q7*, *H2-T10* and *H2-T22* [158]. H2-M3 was also found to harbor potential NLRC5 binding sites. These data corroborate the increased susceptibility of NLRC5 deficient T cells to increased NK cell mediated killing during viral infections [173]. Whether NLRC5 deficiency affects MHC-Ib expression in other tissues and cells, and how this would impact NK cell-mediated tumor killing and the development of adaptive antitumor immunity, are questions that need to be addressed. Analysis of the TCGA dataset revealed that the expression of *HLA-E*, -*F* and -*G* show very strong positive correlation with *NLRC5* and *CD8A* similar to their association with classical MHC-Ia genes. It can be envisaged that NLRC5-deficient clones arising during cancer progression, lacking both MHC-Ia and MHC-Ib, would impair conventional and unconventional CD8^+^ T cell activation towards these clones but would favor NK cell activation. Distinguishing the impact of NLRC5-driven MHC-Ib expression from that of MHC-Ia expression and their contribution to cancer immune surveillance in human tissues will be very challenging. Studies on NLRC5 deficient mice and single cell RNAseq data from human cancers my shed light on this issue.

## 16. Exploiting NLRC5 for Cancer Immunotherapy

### 16.1. Restoring Cancer Immunogenicity

The most obvious and direct application of NLRC5 to cancer immunotherapy would be to restore MHC-I expression in poorly immunogenic cancers (Figure 4). This approach will be beneficial in cancers displaying soft epigenetic alterations of MHC-I or MHC-I pathway genes that harbor the SXY module in their promoters. This can also be applicable to cancers that develop unresponsiveness to IFNs but will not be useful for cancers bearing ‘hard’ lesions of the MHC-I pathway genes such as deletion and nonsense mutations. Using NLRC5 to restore MHC-I will also circumvent the undesired side effect of using type-I and type-II IFN as they also induce the immune checkpoint blocker PD-L1 [226,229,269].

As discussed previously, using the inhibitors of enzymes involved in mediating epigenetic repression of NLRC5 via DNA methylation, histone methylation and histone deacetylation could be a feasible approach (Figure 4), but the off-target effects of these drugs may limit their use. This limitation can be mitigated by selective expression of NLRC5 in cancer tissues. However, unlike small molecule drugs that can be administered orally or parenterally, NLRC5 must be directly delivered to cancer cells and efficient delivery methods need to be developed. Delivery of NLRC5 via oncolytic viruses could be one approach that can be tested as the many OV platforms are in advanced clinical trials for cancer immunotherapy [243,244]. However, oncolytic viruses exploit the loss of IFN-I signaling in cancer cells to specifically target them while sparing normal cells, and therefore their use will be limited in cancers that retain intact IFN-I signaling pathway and antiviral mechanisms. Even though arming OV with NLRC5 is a feasible approach that can achieve both tumor cell killing and elicit effective antitumor immunity, a potential caveat of this approach would be cancer cell lysis may precede effective NLRC5-mediated increase in MHC-I expression and tumor antigen presentation. An alternate and viable approach would be mRNA delivery systems [270]. Indeed, delivery of cytokines and co-stimulators have been shown to increase antitumor immune responses [271,272,273]. It may not be possible to deliver NLRC5 to every cell in the tumor mass. However, immune response elicited by NLRC5-expressing tumor cells can be effective against parental cells bearing low MHC-I [29]. Delivery of NLRC5 to cancers may also be combined with a limited use of epigenetic modifiers to relieve repression on MHC-I and APM promoters.

An important impediment to deliver NLRC5 to cancer cells would be its large size. NLRC5 is a large 205 kDa protein encoded by 49 exons [135], which puts constraints on accommodating the cDNA or RNA into any delivery vehicle. In this context, the Kufer lab has shown that a chimeric NLRC5 containing the NACHT and 4 LRRs of CIITA instead of its own 20 LRRs (Figure 2C,D) retained the ability to transactivate MHC-I genes [274]. Development of minimal NLRC5 constructs retaining the key functionality of transactivating MHC-I and APM genes, and the ability to induce antitumor immunity is a crucial step towards exploiting NLRC5 for intra-tumoral gene delivery.

### 16.2. Identification of Immunogenic Peptides

Another potential application of NLRC5 in cancer immunotherapy is its possible use for the discovery of immunogenic tumor antigenic peptides. Current approaches of tumor antigenic peptide discovery are geared towards using the genomic data to detect cancer-associated mutations within coding sequences and to predict whether the peptide sequence bearing the altered amino acid would bind MHC-I [55,56,57,275]. Even though this approach has led to the identification of immunogenic tumor antigenic peptides, it may not be robust enough to detect all possible tumor antigenic peptides that may be important for tumor immunosurveillance. This consideration is particularly important given the fact that a significant proportion of the MHC-I bound peptides arise from non-linear peptide sequences arising from alternate splicing of mRNA, use of alternate start codons and proteasome-mediated peptide splicing and peptides arising from non-coding sequences [58,59,60,61,62,63,64,65]. Even though the relative importance of these alternate peptides in antitumor immunity is unclear, their possible role in cancer immune surveillance cannot be ignored. Therefore, direct identification of MHC-I-associated peptides (MAPs) from cancer cells compared to normal tissues would be informative. Unfortunately, identification of these peptides is hampered by the lack of reference databases to predict the genomic origin of these peptides. To overcome this limitation, the Perreault laboratory constructed a database of all possible protein sequences in an Epstein-Barr virus-transformed B cell line from RNAseq data that was translated in silico in all three reading frames in forward and reverse directions [60]. This study showed that noncoding genomic sequences and out-of-frame translation of exonic sequences contribute to nearly 10% of MAPs. A similar proteogenomic approach was used to discover novel TSA in high-grade serous ovarian cancer [62]. Even though it would be an enormous undertaking to generate such databases for cancers from every tissue type, an international consortium similar to TCGA can help create such a database in the foreseeable future. Cancer neoantigens discovered through these approaches can be used to induce a TSA-specific immune response, expand TSA-specific CTLs for ACT or generate TSA-specific CAR-T cells. In this endeavor, NLRC5 could be used to facilitate direct identification of MHC-I bound peptides from cancers that display low level of MHC-I expression. Cancer cell lines representative of different cancer types and primary cancer specimens transfected or treated with NLRC5 coding nucleic acids will not only upregulate MHC-I expression but also APM genes that will facilitate processing and presentation of tumor antigenic peptides to boost the power of proteogenomic cancer neoantigen discovery.

### 16.3. Biomarker to Predict Responsiveness to Immune Checkpoint Therapy

Another possible application of the current knowledge on NLRC5 would be to use NLRC5 expression as a biomarker to predict responsiveness to immune checkpoint inhibitor (ICI) therapy. Even though ICI therapy have revolutionized cancer immunotherapy with a low relapse rate and longer disease-free survival, it is effective only in a subset (~20%) of cancer patients [276,277,278]. Diverse factors, including an intestinal microbiome, may contribute to poor responsiveness to ICI [279,280,281]. Biomarkers that can reliably predict responsiveness to ICI therapy are not yet available [282]. As it is unlikely that a single biomarker would have the predictive power, efforts are being made to develop a scoring system using multiple parameters [283,284,285]. In parallel, combinatorial approaches using a second checkpoint inhibitor, radiation, chemotherapy, ACT using CAR-T cells and oncolytic virus therapy are being investigated to improve the outcome of ICI therapy [286,287,288,289]. However, the success of all these approaches relies on the expression of MHC-I:tumor Ag peptide on tumor cells. Obviously, tumors that repress MHC-I or APM will not efficiently induce antitumor T cells and will also evade CTLs, diminishing the effectiveness of ICI therapy. Expression of MHC-I itself could be one of the biomarkers to predict unresponsive cases. However, as discussed before, MHC-I defect could arise from hard/irreversible or soft/reversible genetic lesions, and the latter may include impaired NLRC5 expression. As *NLRC5* is the most frequently affected MHC-I pathway gene in many human cancers, and NLRC5 expression positively correlates with patient survival in several cancers including melanoma, cervical cancer and bladder cancer, Yoshihama et al., proposed NLRC5 as a potential prognostic biomarker for tumor immune evasion [28]. In fact, identification of NLRC5 deficient cancers would also make it possible to identify patients who are likely to benefit from therapeutic restoration of NLRC5 (discussed in an earlier section) that can boost the effectiveness of ICI therapy and generate lasting antitumor immunity.

To use NLRC5 as a biomarker, appropriate reagents and methods need to be developed and validated. PCR based evaluation of NLRC5 gene expression in tumor biopsies would not be appropriate as NLRC5 is highly expressed in hematopoietic cells that infiltrate tumors. Immunohistochemical (IHC) detection of NLRC5, along with other markers routinely used in clinical oncology such as Ki67 and HER2 would be ideal. However, antibodies used for clinical diagnosis generally go through extensive validation in multiple institutions before being approved for diagnostic purpose [290,291]. Validation of the currently available monoclonal antibody generated by the Kufer laboratory [137] (clone 3H8, commercially available from MilliporeSigma), and generation and validation of other antibodies would be a step forward in this direction. A few studies have reported elevated NLRC5 protein expression in certain cancers that is associated with poor survival [216,292,293]. The IHC staining of NLRC5 in these studies was less than clear and the proposed mechanistic underpinnings of such negative correlations were tenuous. Developing optimized reagents and methods will clarify such nuances on the predictive potential of NLRC5 expression in cancers.

## 17. Conclusions

Since its discovery as the transcriptional coactivator of MHC-I and certain APM genes in 2010, NLRC5 has raised the hope of exploiting it to correct MHC-I expression defects in cancer immunity. This idea gained traction with the findings that the loss of NLRC5 expression resulting from genetic and epigenetic causes is the most common mechanisms of MHC-I deficiency in cancers, and that loss of NLRC5 has poor prognosis for many cancers. NLRC5 can be used not only to restore MHC-I expression and tumor immunogenicity but also to discover cancer antigenic peptides and to predict unresponsiveness to cancer immunotherapy. However, certain limitations of using NLRC5 must be overcome to realize the translational potential of NLRC5. Nonetheless, recent advances in understanding the epigenetic regulation of NLRC5 has opened alternate approaches to restore MHC-I expression in cancers. There are also several outstanding questions on the regulation of NLRC5 expression and functions that are elaborated throughout the review. The key questions are summarized below.

## 18. Outstanding Questions

1. What is the basis of differential NLRC5 expression in various non-hematopoietic tissues? Different studies on NLRC5 deficient mice have reported varying levels of NLRC5 mRNA. Given that NLRC5 expression is modulated by IFNs and TLRs, and NLRC5 is also regulated at post-transcriptional level by miRNAs and lncRNAs, evaluation of NLRC5 mRNA and protein levels in normal mouse and human tissues will be informative. Whether the NLRC5 isoforms described in human leukocytes are also expressed in somatic cells also needs to be addressed.

2. Is the epigenetic control of NLRC5 expression differentially regulated in different cell types? EZH2-mediated histone methylation represses NLRC5 in stem cells and cancer cells, whereas IFNγ promotes the same events in macrophages on certain genes. As IFNγ strongly induces NLRC5 gene expression, it will be informative to dissect the impact of STAT1-mediated transcriptional activation on different modes of epigenetic repression of *NLRC5*.

3. Why does the predictive potential of NLRC5 expression vary in different cancers? In TCGA study cohorts, not all cancers show reduced NLRC5 expression and low NLRC5 expression is not associated with poor prognosis in all cancers. In fact, liver, colon and brain cancer tissues show elevated NLRC5 expression, which is thought to result from high inflammatory conditions. Some studies even reported that high NLRC5 is associated with poor prognosis, especially in HCC. As the TCGA datasets represent only bulk mRNA from cancer, stromal and immune cells, additional parameters such as immune and stromal cell signatures could be used to stratify the TCGA study cohorts and evaluate the predictive potential of NLRC5 expression in cancer subsets.

4. Is the predictive potential of NLRC5 influenced by the mutational load and the frequency of neoantigen generation of different cancers?

5. Is NLRC5 essential or dispensable for tumor immune surveillance? Even though NLRC5 overexpression enhances tumor immunogenicity, whether NLRC5-independent MHC-I expression is sufficient for cancer immunosurveillance has not been yet experimentally addressed.

6. Does MHC-Ib expression in somatic cells require NLRC5? Does this impact NK cell-mediated cancer immune surveillance?

7. Irradiation of cancer cells induces NLRC5 expression. Does it require radiation-induced type-I IFN or not? If it occurs independently then what are the underlying mechanisms? Is the immunogenic cell death caused by irradiation depends on NLRC5? Can radiation induced NLRC5 can be tuned to allow for improved antigen presentation before tumor cell death?

8. To what extent the size of NLRC5 required for its transactivation of MHC-I and APM genes be minimized to facilitate its translational potential for cancer immunotherapy and cancer antigen discovery?

## Figures and Tables

**Figure 1 ijms-22-01964-f001:**
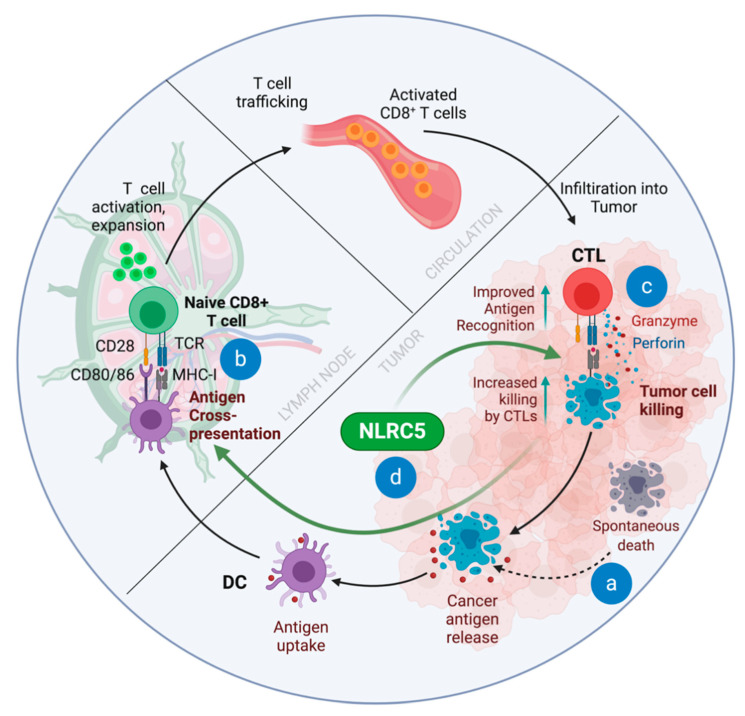
Cancer-Immunity cycle and the points of intervention for NLRC5. Spontaneous death of cancer cells leads to the release of cancer antigens (**a**), which are taken up by dendritic cells and presented via cross-presentation to naïve CD8^+^ T cells in draining lymph nodes (**b**). Activated CD8^+^ T cells undergo clonal expansion, differentiate to effector CTLs, enter circulation and traffic to the tumor. Upon recognizing the antigenic peptide on cancer cells, CTLs release their cytotoxic granules to cause tumor cell killing (**c**). Release of more tumor antigens and their cross-presentation to additional naïve CD8^+^ T cells results in reiteration of this cycle leading to tumor elimination. However, this process can be impaired at multiple nodes of this cancer immunity cycle including immune checkpoint inhibition at CTL-mediated killing and naïve T cell activation. Tumors that downmodulate MHC-I escape immune detection and killing, thereby dampening the development and execution of antitumor immunity. As genetic lesions and epigenetic modifications of the *NLRC5* gene are the most common causes of MHC-I downregulation in cancers, restoration of NLRC5 (**d**) will restore tumor immunogenicity, leading to efficient killing of tumor cells and potentiation of antitumor immunity (thick green arrows).

**Figure 2 ijms-22-01964-f002:**
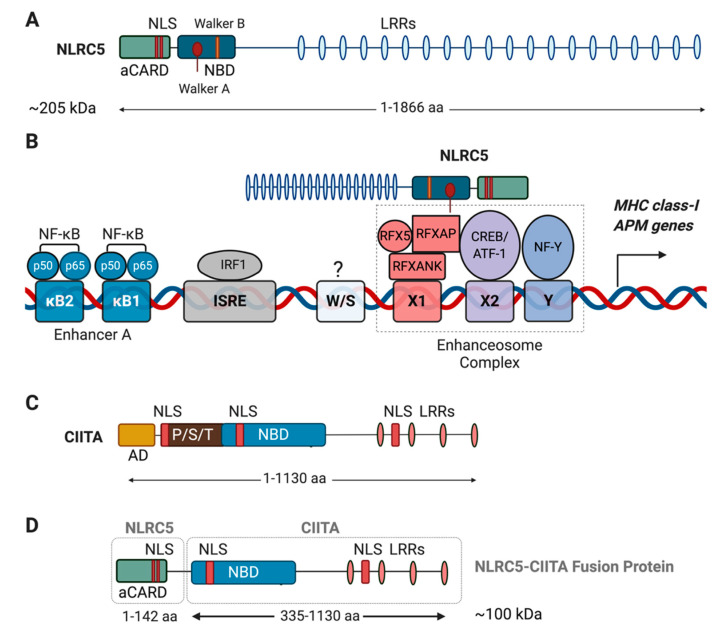
Structure and transactivation functions of NLRC5. (**A**) NLRC5 (1866 aa) displays a tri-partite structure consisting of (i) an atypical CARD (aCARD; 1–139 aa) domain with a helix extending to residue 161 and a bipartite Nuclear Localization Sequence (NLS; 121–122, 132–134 aa), (ii) a Nuclear Binding Domain (NBD; 197–369 aa) that harbors Walker A and Walker B motifs (228–235, 303–313 aa) and (iii) twenty Leucine Rich Repeats (LRRs) (589–1866 aa). (**B**) NLRC5 transactivates MHC-I and APM genes via the SXY module. NLRC5 lacks a DNA binding domain and thus interacts through the enhanceosome complex formed by the transcription factors (RFX proteins, CREB/ATF1, NF-Y) bound to X1, X2 and Y box motifs. RFX5 acts as a key mediator for binding of NLRC5 with the promoter. The LRRs of NLRC5 are compressed for clarity. The MHC-I promoters also harbor an interferon stimulated responsive element (ISRE) and κB consensus sites that bind IRF1 and NF-κB, respectively. (**C**) Structure of CIITA (NLRA; 1130 aa). CIITA has an acidic activation domain (AD) (1–125 aa), three nuclear localization sequences (141–159, 405–414, 955–959 aa), a P/S/T domain (Pro/Ser/Thr; 126-322 aa), an NBD (336–702 aa) and four LRRs (930–1130 aa). (**D**) The Kufer laboratory has engineered an NLRC5-CIITA fusion protein containing the aCARD domain of NLRC5 and NBD and LRRs of CIITA. This fusion protein transactivates MHC-I genes as efficiently as the full length NLRC5.

**Figure 3 ijms-22-01964-f003:**
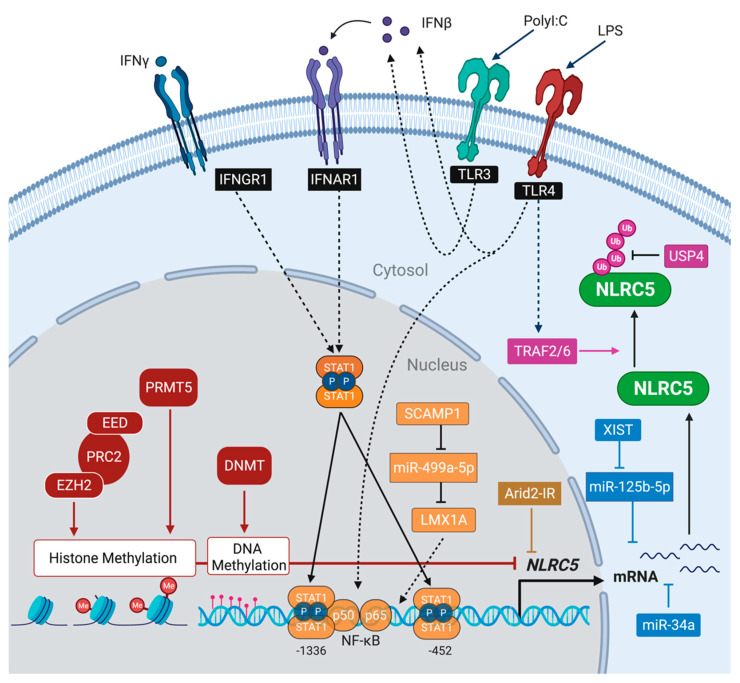
Regulation of *NLRC5* gene expression. IFNγ strongly induces *NLRC5* gene expression through transcriptional activation of STAT1. STAT1 homodimers bind the GAS sequences in the *NLRC5* promoter (-452, -1336). Type-I IFNs, induced by TLR ligands also induce NLRC5 via STAT1 activation. NF-κB activation by TLRs can synergize with STAT dimers by binding to the distal GAS site. NLRC5 is regulated by (i) epigenetic modifications of its promoter as well as at the (ii) transcriptional, (iii) post transcriptional and (iv) post-translational levels as detailed below: (**i, factors indicated in burgundy color**) Epigenetic regulation. NLRC5 promoter may be repressed by DNA methylation by DNA methyl transferases (DNMT), histone lysine methylation by the polycomb repressive complex 2 (PRC2) (containing lysine methyltransferase EZH2 [Enhancer of Zeste Homolog 2] and EED [Embryonic Ectoderm Development]) or histone lysine methylation by protein arginine methyltransferase 5 (PRMT5). (**ii, orange**) Transcriptional activators (STAT1 and NF-κB) and their modulators. The LncRNA Arid2-IR (AT-rich interactive domain 2-intronic region) binds to the *NLRC5* promoter and prevents its transcription. The lncRNA SCAMP1 (Secretory carrier-associated membrane protein 1) modulates NLRC5 transcription by removing miRNA miR499-5p, thereby relieving its repressive effect on LMX1A (LIM homeobox transcription factor 1) that binds to multiple sites at the *NLRC5* promoter. (**iii, brown, blue**) Post-transcriptional regulation by miRNAs. miR-34a and miR-125b-5p downregulate NLRC5 expression by destabilizing mRNA and inhibiting translation. miR-125b-5p is regulated by the lncRNA XIST. (**iv, purple**) Post-translation regulation. NLRC5 is ubiquitinated by the TRAF2/6 complex induced by TLR signaling. Ubiquitin-specific protease 14 (USP14) deubiquitinates NLRC5.

**Figure 4 ijms-22-01964-f004:**
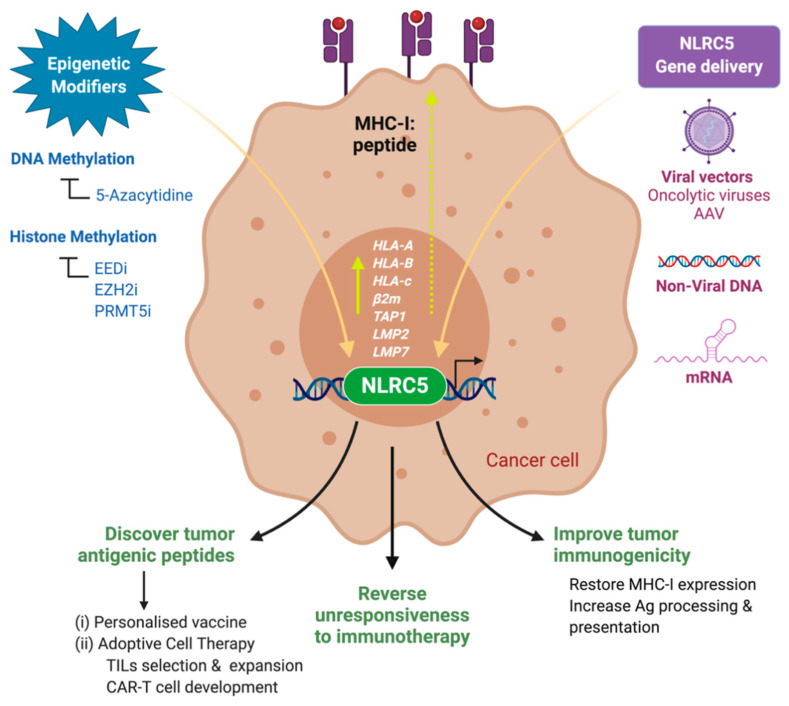
Strategies to restore NLRC5 expression in cancer cells and its potential applications for cancer immunotherapy. MHC-I low cancers can be treated with pharmacological agents to relieve epigenetic repression of *NLRC5* that will also derepress *MHC-I* and APM genes. NLRC5 gene delivery using oncolytic viruses, naked DNA or mRNA will be useful on cancers with a dysfunctional NLRC5 gene. Restoration of NLRC5 can be exploited to improve tumor immunogenicity and CTL-mediated killing, and to reverse unresponsiveness to conventional cancer immunotherapy such as immune checkpoint inhibitors. NLRC5 can also be exploited for cancer neoantigen discovery that will find application in personalized cancer vaccines and adoptive cell therapy.

**Table 1 ijms-22-01964-t001:** Studies ascribing antitumor or protumor roles to NLRC5.

Cancer Type	Model Systems	Study Description	Molecular Effects	Ref.
**NLRC5 as a tumor suppressor**				
Melanoma	Murine B16.F10 cell line stably expressing NLRC5	NLRC5 limits tumor growth and metastasis in C57BL/6 mice by activating antitumor CD8^+^ T lymphocytes;	NLRC5 upregulates MHC-I, β2M, PSMB9, PSMB8, TAP1 gene expression; NLRC5 promotes presentation of peptide from gp100 (Pmel-1) tumor antigen	[208]
Melanoma	PRMT5 knockdown in B16.F10 and Yummer1.7 cell lines; B16.F10 stably expressing NLRC5	Induction of endogenous NLRC5 by PRMT5 knockdown, or stable NLRC5 expression inhibited tumor growth in C57BL/6 mice	PRMT5 reduces NLRC5 expression by promoting methylation of Arg residues on histones	[176]
Melanoma	Jak1^−/−^B16 cell line expressing NLRC5	NLRC5 rendered Jak1^−/−^ B16 cells susceptible to killing by adoptively transferred Pmel-1 TCR transgenic CD8^+^ T cells in vivo	NLRC5 upregulates MHC-I	[210]
Pancreatic adeno-carcinoma (PDAC)	Murine Panc02 cell line expressing a model antigenic peptide SIYRYYGL fused to GFP (Panc02SIY100)	Gamma irradiation induces NLRC5 expression renders Panc02 cells susceptible to anti-PD-L1 in vivo; Stable NLRC5 expression in Panc02SIY100 promotes activation of 2C TCR transgenic CD8^+^ T lymphocytes	Gamma irradiation induces MHC-I independently of IFN-I signaling	[178]
**NLRC5 as a tumor promoter**				
Hepatocellular carcinoma (HCC)	Human HCC specimens; HCC cell lines: HepG2, SMMC-7721, BEL-7402; stable expression or knockdown of NLRC5	HCC specimens and cell lines display elevated NLRC5 expression; NLRC5 promotes cell proliferation, migration and invasion; NLRC5 knockdown has opposite effect & reduces HepG2 tumor growth in nude mice	NLRC5 expression promotes Wnt/β-catenin signaling and c-Myc, CyclinD1, MMP3 expression; β-catenin inhibitor iCRT3 attenuated the effects of NLRC5 overexpression	[211]
Hepatocellular carcinoma	HepG2 cell line	NLRC5 overexpression in HepG2 cells promotes cell growth via upregulating VEGF-A	NLRC5 promotes VEGF-A expression via AKT activation	[213]
Clear cell renal cell carcinoma (ccRCC)	Human ccRCC specimens Human ccRCC cell lines Caki-1, 786-O and 769-P	ccRCC specimens and cell lines display elevated NLRC5 expression; NLRC5 promotes cell proliferation, migration and invasion; NLRC5 knockdown causes opposite effects and reduces 786-O tumor growth in nude mice	NLRC5 promotes β-catenin, c-Myc, CyclinD1, MMP2, MMP9 expression	[212]
Giloma	Human glioma tissues, cell lines U87, U251	High grade glioma tissues and cell lines display elevated NLRC5 expression due to high levels of lncRNA SCAMP1; NLRC5 knockdown restrains cell proliferation, migration & invasion and increases apoptosis	SCAM1-1 sponges off miR-499a-5p, which targets LMX1A; LM1A binds NLRC5 promoter and is proposed to augment NLRC5 expression to activate the Wnt/β-catenin pathway	[192]
Esophageal squamous cell carcinoma (ESCC)	ESCC cell lines	NLRC5 overexpression in ESCC cell lines promotes cell proliferation, colony formation and cell cycle progression	miR-4319 targets NLRC5; low miR-4319 in ESCC upregulates NLRC5 expression	[215]
Breast cancer	Breast cancer tissues, cell lines MCF-7, MDA-MB-231	MCF-7 and MDA-MB-231 show increased proliferation, migration and invasion due to elevated expression of XIST lncRNA, which upregulates NLRC5	XIST sponges off miR-125b-5p, which targets NLRC5	[197]

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
