# Peer review of "The MHC Class-I Transactivator NLRC5: Implications to Cancer Immunology and Potential Applications to Cancer Immunotherapy"

_ijms, 2021, doi:10.3390/ijms22041964_

Round 1

Reviewer 1 Report

In this review, Shukla and colleagues give an overview of cancer immunity mediated by CD8T cells and the implications of NLRC5 in the MHC-I pathway.

The review is well written, gives a comprehensive overview on cancer immunity and the role of NLRC5 both as trans-activator of MHC-I expression and its implication in boosting cancer immuno-therapy. The latest discoveries in the filed are mentioned and taken into consideration to discuss openly the potential to use NLRC5 to render low immunogenic tumors more visible to the immune system (by increasing its expression and consequently the level of expression of MHC-I). A full section of open questions, that need to be still addressed and that could help scientist in advancing their research is included and nicely structure. 

However there are few minor points that authors should take into considerations before publications. 

  1. The authors give an extensive introduction, in the first section of the review, to the topic, mainly on cancer immunity and they introduce NLRC5 at the very end. Despite the extensive description of NLRC5's functions, regulation and involvement in cancer immunity later on in the review, considering the title of the manuscript, I would have expected to read some words more about NLRC5 in this section. I suggest the revised the end of this section, putting a bit more emphasis on NLRC5.
  2. In the section 4 "The cancer-Immunity cycle and 'immune visibility' of cancers", none of the reference 118-121 describe the K63 post-translational modification as a mechanism for MHC-I degradation. All the papers mentioned here describe the importance of ubiquitination of MHC-I for its internalization but there is no evidence of K63 chains. I would instead refer to: "Journal of Immunology 2010, Duncan LM, et al" or "Traffic 2010, Boname JM, et al". I therefore suggest to update the bibliography for this part. 
  3. The authors introduce in the section 6 " Loss of NLRC5 expression frequently underlies reduced MHC-I expression in cancers", the NLR receptor family as Pattern Recognition Receptors (PRRs) and compare them just to TLRs. Despite the correct information given, the cytosolic and organelle bound PRRs count more than just TLRs and NLRs. I suggest here either to remove the comparison to the TLRs (since is it not necessary for the scope of the review) or if the authors want to keep this info to mention also the others PRRs for the sake of clarity for the readers.  Moreover a reference at line 286 for the role of NOD1, NOD2 or NLRP3, NLRC4 is missing. Please provide a reference here.  I would also encourage the authors, since they mention the importance of NLRC5 in regulating NFκB signalling, to discuss the controversy about this function. Few words would be sufficient but would give the reader the complete picture. 
  4. In the section 5 " Structure and transcriptional coactivator function of NLRC5", I would give a bit more details about the specificity of NLRC5 for MHC-I transactivation and the importance of the S-box in the promoter of the target genes. It is well known that CIITA is more promiscuous than NLRC5 (over-expression of CIITA leads to transactivation of also MHC-I) and few words more on this specificity would stress more the importance of NLRC5 in regulating MHC-I. 
  5. In the section 15 "Role of NLRC5 in Cancer immune surveillance" the authors discuss the role of NLRC5 in regulating canonical and non-canonical MHC-I genes. I found a bit confusing the description on NK cells, their inhibition and activation related to NKGA and MHC. Could authors be more clear in explaining this concept and underlying the importance for cancer-immunity?
  6. General comment: there are several typing mistakes that need to be correct both in the main text and in the figure legends. Line 192 (figure legend 2): ....can be impaired at multiple?? Line 875 (OR and not OF). The same mistake is present at line 962.

In conclusion, the review is well structured and gives a detailed overview on the current knowledge of cancer immunity, the role of NLRC5 as trans-activator of MHC-I expression, its correlation with low immunogenic tumors and the potential of using it as booster of MHC-I expression to render tumors more visible to the immune system. A discussion about the possible strategies on how to boost NLRC5's expression in cancer cells and their consequences is present and nicely described. 

Reviewer 2 Report

This manuscript is well-written review covering all aspect of NLRC5 in relation to cancer immunology and immunotherapy including issues in NRLC5 research and potential solutions for their elucidation. In the chapter describing defective MHC-I expression in cancers, the effect of Fhit on MHC-I expression could be mentioned as Fhit can also be used (similarly to NLRC5) for MHC-I upregulation in tumors (Cancers 2020; 12(6): 1563).

Minor comments:

Some inconsistencies could be corrected, e.g. anti-tumor vs. antitumor, MCF7 vs. MCF-7, PMEL-1 vs. Pmel-1, B2M vs. β2M.

In Figure 3 legend, colors are assigned repeatedly (brown color for both post transcriptional (line 509) and transcriptional regulation (line 516)).

Table 1 is a bit confused.

Line 82: “killing by oncolytic virus“ is one example of “inducing immunogenic cell death of tumors“.

Line 83: Can you specify “their“?

Line 192: Can you specify “multiple“?

Line 245: Impairment of the B2M gene usually results in the total HLA-Ia loss.

Line 249: As mouse models are also mentioned, “HLA-I” should be replaced with “MHC-I”.

Line 292: “NK-“ should be replaced with “NF-“.

Line 529: … methylation and? causes…

Line 667: “Nlcr5“ should be replaced with “Nlrc5“.

Line 694: Studies “in mice“ are not “in vitro studies”.

Lines 738-739: GAS sequences are present in IRFs and NLRC5 promoters.

Lines 755-756: OV can also be armed with genes encoding immunostimulatory molecules …

Line 875: “of MHC-I of MHC-I“ should be corrected.

Line 967: “make it to possible identify“ should be corrected.

Line 979:  “MilliporeSigma” should be corrected.
